# Pharmacokinetics of Antimicrobials in Children with Emphasis on Challenges Faced by Low and Middle Income Countries, a Clinical Review

**DOI:** 10.3390/antibiotics12010017

**Published:** 2022-12-22

**Authors:** Kevin Meesters, Tinsae Alemayehu, Sofia Benou, Danilo Buonsenso, Eric H. Decloedt, Veshni Pillay-Fuentes Lorente, Kevin J. Downes, Karel Allegaert

**Affiliations:** 1Department of Pediatrics, BC Children’s Hospital and The University of British Columbia, 4500 Oak Street, Vancouver, BC V6H 3N1, Canada; 2Division of Pediatric Infectious Diseases, Department of Pediatrics and Child Health, St. Paul’s Hospital Millennium Medical College, Addis Ababa P.O. Box 1271, Ethiopia; 3Division of Infectious Diseases and Travel Medicine, American Medical Center, Addis Ababa P.O. Box 62706, Ethiopia; 4Department of Pediatrics, General University Hospital of Patras, Medical School, University of Patras, 26504 Rion, Greece; 5Department of Woman and Child Health and Public Health, Fondazione Policlinico Universitario Agostino Gemelli IRCCS, Università Cattolica del Sacro Cuore, Largo A. Gemelli 8, 00168 Rome, Italy; 6Centro di Salute Globale, Università Cattolica del Sacro Cuore, 00168 Rome, Italy; 7Division of Clinical Pharmacology, Department of Medicine, Faculty of Medicine and Health Sciences, Stellenbosch University, Cape Town 7500, South Africa; 8Department of Pediatrics, Perelman School of Medicine, University of Pennsylvania, 3400 Civic Center Blvd, Philadelphia, PA 19104, USA; 9Division of Infectious Diseases, The Children’s Hospital of Philadelphia, 3401 Civic Center Blvd, Philadelphia, PA 19104, USA; 10Department of Development and Regeneration, KU Leuven, Herestraat 49, B-3000 Leuven, Belgium; 11Department of Pharmaceutical and Pharmacological Sciences, KU Leuven, Herestraat 49, B-3000 Leuven, Belgium; 12Department of Clinical Pharmacy, Erasmus Medical Center, Doctor Molewaterplein 40, 3015 GD Rotterdam, The Netherlands

**Keywords:** antimicrobials, children, pharmacokinetics, developmental pharmacology, PK/PD targets

## Abstract

Effective antimicrobial exposure is essential to treat infections and prevent antimicrobial resistance, both being major public health problems in low and middle income countries (LMIC). Delivery of drug concentrations to the target site is governed by dose and pharmacokinetic processes (absorption, distribution, metabolism and excretion). However, specific data on the pharmacokinetics of antimicrobials in children living in LMIC settings are scarce. Additionally, there are significant logistical constraints to therapeutic drug monitoring that further emphasize the importance of understanding pharmacokinetics and dosing in LMIC. Both malnutrition and diarrheal disease reduce the extent of enteral absorption. Multiple antiretrovirals and antimycobacterial agents, commonly used by children in low resource settings, have potential interactions with other antimicrobials. Hypoalbuminemia, which may be the result of malnutrition, nephrotic syndrome or liver failure, increases the unbound concentrations of protein bound drugs that may therefore be eliminated faster. Kidney function develops rapidly during the first years of life and different inflammatory processes commonly augment renal clearance in febrile children, potentially resulting in subtherapeutic drug concentrations if doses are not adapted. Using a narrative review approach, we outline the effects of growth, maturation and comorbidities on maturational and disease specific effects on pharmacokinetics in children in LMIC.

## 1. Introduction

Despite widespread implementation of childhood immunization programs, improvements in hygiene and better access to healthcare, infections have remained the most frequent cause of childhood mortality in low and middle income countries (LMIC) [1,2]. Antimicrobials are potentially lifesaving and are among the most frequently administered drugs in children. Unfortunately, the development of antimicrobial resistance (AMR) is a direct consequence of antimicrobial use. LMIC are disproportionately affected by AMR [3,4], driven by numerous circumstances such as excessive use of broad spectrum antimicrobials in healthcare settings, supply issues, limited availability of diagnostic resources, and over-the-counter dispensation [5]. Antimicrobial stewardship (AMS) is a coordinated set of actions to preserve antimicrobials through enhancing the quality of antimicrobial prescribing and has been proposed as a tool in to combat AMR [6]. While unnecessary antimicrobial prescription should be avoided [7], adequate dosing is also of utmost importance since subtherapeutic concentrations result in therapy failure and selection of drug-resistant organisms [8,9]. Achievement of the effective concentration of a drug is governed by the pharmacokinetic processes absorption, distribution, metabolism and elimination, which is depicted in Figure 1.

The need for adapted drug therapies for children has been increasingly acknowledged, and pediatric pharmacokinetic studies have expanded during the most recent years to optimize dosing in children. A good understanding of pharmacokinetics of antimicrobials is critical for successful treatment of infections and to combat AMR. In this clinical review, we outline the key processes that dictate the pharmacokinetics of antimicrobials to treat bacterial infections in children, detailing important considerations for antimicrobial dosing, as well as highlighting specific challenges in LMIC.

## 2. Pharmacokinetic–Pharmacodynamic Interaction of Antimicrobials

For each bacterium, the lowest concentration at which an antimicrobial inhibits growth is defined as the minimum inhibitory concentration (MIC) [10]. Since antimicrobials target bacteria, MIC is a key component determining whether the concentration achieved at the site of infection will lead to treatment success or failure. The relationship between the free (unbound) concentration of an antimicrobial over time and the bacterial killing effect *in vitro*, is described by pharmacokinetic-pharmacodynamic (PK/PD) indices (Figure 2). Broadly speaking, antimicrobials can be categorized as concentration-dependent or time-dependent agents [11]. The bacterial killing effect of concentration-dependent antimicrobials is characterized by the ratio between the peak concentration (C_max_) and MIC. Hence, this is directly related to the maximum concentrations that can be achieved at the site of infection. Although bacterial killing of concentration-dependent drugs is dose-dependent, dosing is curtailed by the minimum concentration at which toxicity usually occurs (minimum toxic concentration; MTC); the area between MIC and MTC is called the therapeutic window [12]. Concentration-dependent antimicrobials also tend to exert a post-antimicrobial effect, thereby maintaining a significant antimicrobial effect below the MIC. On the contrary, time-dependent antimicrobials kill bacteria for as long as concentrations are above the MIC. Therefore, the time above the MIC (T > MIC) is the PK-PD index best associated with efficacy for these antimicrobials [13]. Last, the bacterial killing effect of some antimicrobials can be described as being both concentration- and time-dependent; the ratio between area under the concentration-time curve and MIC (AUC/MIC) is the index of choice for concentration-dependent with time-dependent antimicrobials [13].

## 3. Pharmacokinetics: Drug Transport through Cell Membranes

Antimicrobials are mostly administered distant to their site of action and therefore require transportation to the target site. Drugs that are administered via an extravascular route (e.g., enteral, transdermal, intramuscular) must be absorbed into the bloodstream for distribution to target sites. Once in the circulation, drugs distribute rapidly to the organs to which blood flows instantaneously such as the heart, liver and kidneys; this is called the central compartment [14]. Subsequently, some drugs may distribute to the peripheral compartment, which refers to organs, tissues and cells that are perfused at a slower rate. In order to egress the intravascular space, a drug molecule must cross through cell membranes, either via passive diffusion or using active cell processes that involve receptors, transport proteins or transcytosis [15]. Cell membranes are highly hydrophobic and negatively charged. The drug’s ability to transport across cell membranes is governed by intrinsic physicochemical properties of the drug, such as the amount of protein binding, molecular size, charge, lipophilicity and partition coefficient [16]. Drugs that are bound to a protein, either intravascularly or in the extravascular space, cannot move across membranes due to its molecular size [17]. At any body site, drugs reach an equilibrium between the amount that is either protein bound or free. Albumin, the main drug binding protein, is alkalic and therefore tends to bind to acidic drugs. Generally, lipophilic and uncharged molecules diffuse easily across membranes, whereas hydrophobic and charged molecules are dependent on active processes to disseminate into the peripheral compartment [12]. LogP, the logarithm of the partition coefficient, expresses the affinity of a molecule to dissolve in either water or octanol, and hence the lipophilicity of a drug. In general, a LogP less than 0 indicates a hydrophilic molecule, whereas LogP greater than 0 reflects a lipophilic molecule [18]. The degree of ionization of a drug is predicted by the acid-base dissociation constant (pK_a_) that relates the pH at which the ionized and unionized forms exist in equal amounts [19]. Therefore, the amount of unionized drug molecules is not constant, since pH varies in different cells and tissues. Table 1 outlines the physicochemical characteristics of antimicrobials.

## 4. Pharmacokinetic Processes

### 4.1. Absorption

Absorption, the transfer of a drug from the site of administration into the bloodstream, is relevant for all drugs that are not injected intravascularly. Bioavailability (F), the extent to which a drug enters the circulation following administration, is by definition 100% when the drug is infused directly into the bloodstream, but lower for all other routes of administration [59].

The oral route is the most common route of administration for antimicrobials. Following oral ingestion, the drug needs to dissolve into smaller particles that can subsequently be absorbed [60]. Since liquid formulations are better dissolved than tablets within the GI tract, liquids have typically greater bioavailability than tablets [61]. In addition to molecular characteristics of a drug, different physiologic factors determine the amount of GI absorption, including the pH at the different parts of the GI tract, gastric emptying, intestinal motility and perfusion [62]. Furthermore, directly after intestinal absorption, some drugs undergo first-pass metabolism in the portal circulation, before reaching the systematic circulation. This reduces systemic drug concentrations relative to the amount absorbed of a drug [63].

Within 48 h after birth, gastric pH decreases to around 3, and then gradually returns to neutrality by day 8–10 of life. Thereafter, the gastric pH slowly declines again to reach adult values at about two years of age [62,63]. These pH changes are less apparent in infants born before 32 weeks’ gestation [64]. A smaller fraction of acid-labile antimicrobials (e.g., benzylpenicillin, ampicillin, amoxicillin, nafcillin, flucloxacillin and erythromycin) will become inactivated in a higher intragastric pH, and will therefore attain a higher bioavailability in neonates and infants [65].

Gastric emptying is delayed immediately after birth, but approach adult values within the first six to eight months of life [62,64]. Delayed gastric emptying typically leads to degradation of drug molecules through prolonged exposure to intragastric acid, which impairs drug absorption [66]. Similarly, intestinal motility is also reduced in neonates and young infants. However, this results in increased transit time and improved absorption; intestinal motility gradually increases by 6–8 weeks of life [64]. Last, immaturity of secretion and activity of bile and pancreatic fluid leads to impaired fat digestion in neonates and infants, which particularly reduces the dissolution of lipophilic drugs, and hence impede this absorption [67]. Concluding, gastrointestinal (GI) absorption is complex and highly variable across drugs, particularly in neonates. Contrasting processes lead to large inter-individual variability in absorption rates across infants. Nevertheless, commonly used antimicrobials (amoxicillin, cephalexin, cefpodoxime) often attain adequate serum concentrations to treat most relevant neonatal pathogens (*E. coli*, *S. pneumoniae*, *S agalactiae*) [68].

There is a complex interaction between food and drug absorption. Different food categories affect absorption in varying ways [66]. High fat meals delay gastric emptying and impair absorption of hydrophilic drugs, but improve absorption of lipophilic drugs (e.g., itraconazole) by enhancing solubility. High protein nutrients increase intestinal blood flow and may thereby increase absorption. However, bioavailability of drugs with similar structures to peptides (e.g., cephalexin, cefadroxil) can be lowered if ingested with proteins. High fiber foods delay gastric emptying, reduce solubility of drugs, and decrease bile salt concentrations. Moreover, fasting decreases gastric pH and leads to delayed gastric emptying, while enhancing splanchnic blood flow and stimulating release of bile salts [66]. These complex interactions make it important for clinicians to be aware about when drugs should be taken with food. Table 2 summarizes food effects on commonly used antimicrobials.

Some antimicrobials are administered intramuscularly. Similar physiologic principles determine the bioavailability after intramuscular absorption, such as the local pH and the tissue perfusion rate. As neonates and infants have low muscle mass and low regional blood flow to muscles, bioavailability via intramuscular administration is lower in infants [79].

### 4.2. Distribution

Volume of distribution (V_d_) is a pharmacologic parameter that relates the amount of a drug in the body to its measured concentration in blood or plasma [14]. It is an apparent volume, since it may well exceed any physiologic volume required to contain all the drug in the body at the measured concentration. The magnitude and the sites of drug distribution are dependent on the physicochemical properties of the drug and different biologic factors, such as body composition and various physiological processes, which are all altered by both ontogeny and disease states [80].

Body composition changes markedly throughout childhood. Premature neonates have a much higher total body water content than term born infants. This increases V_d_ of hydrophilic drugs, such as aminoglycosides and glycopeptides. As a result, higher doses per kilogram are necessary at initiation for premature neonates to attain the same target serum concentration, compared to term born infants [81]. For some drugs, loading doses are given to rapidly attain optimal concentrations. Since C_max_ is directly related to V_d_, neonates need higher loading doses of hydrophilic antimicrobials [82]. Furthermore, protein composition evolves in childhood. In infants, lower protein binding has been reported [83]. This increases V_d_ of antimicrobials that are highly protein bound, such as ceftriaxone. Hypoalbuminemia is common in many diseases, such as nephrotic syndrome, liver failure and cachexia. Under these circumstances, the free fraction of protein bound drug rises [84]. However, this does not necessarily translate to higher drug exposure, as unbound molecules are available for excretion, more unbound drug is cleared [85]. Furthermore, free drug molecules may egress the circulation and bind to extravascular proteins.

Tight junctions and multiple cellular mechanisms prevent substances and micro-organisms from entering the brain and cerebrospinal fluid (CSF), which is known as the blood-brain barrier [86]. Therefore, some antimicrobials may not cross the blood-brain barrier at all. But, the integrity of the blood-brain barrier decreases during central nervous system infection (CNS) [87]. In general, unbound lipophilic and uncharged drugs enter the CSF at a higher rate than hydrophilic drugs with a large size or charge. The abilities of antimicrobials to penetrate CSF are displayed in Table 3.

Efflux transporters, such as p-glycoprotein (P-gp), avert intracellular transport of xenobiotics and toxic substrates [88,89]. As these transporters excrete certain drug molecules, this restricts the distribution of some antimicrobials. On top of that, drugs may both induce and inhibit P-gp, such as rifampicin (inducer) and protease inhibitors (inhibitor), and hence further affect exposure to substrates.

**Table 3 antibiotics-12-00017-t003:** Cerebrospinal fluid penetration of antimicrobials.

Agent	Cerebrospinal Fluid (CSF) Penetration
Aminoglycosides	
Amikacin Gentamicin Tobramycin	Systemic amikacin, gentamicin and tobramycin penetrate the CSF of inflamed meninges to a limited extent. Their clinical use for CNS infections is restricted by toxicities if administered intravenously. Intrathecal doses of amikacin, gentamicin, and tobramycin have been reported to be effective and well tolerated [90,91]. No information available for other aminoglycosides.
Antimycobacterials	
Ethambutol	Limited data suggest poor to moderate CSF penetration of inflamed meninges [92].
Isoniazid	CSF concentration comparable with plasma concentration in inflamed meninges [92].
Pyrazinamide	CSF concentration comparable with plasma concentration in inflamed meninges [92].
Rifabutin	Higher CSF penetration than rifampicin, but toxicities may restrict its use in CNS infections [93].]
Rifampicin	Moderate CSF penetration at standard doses, therefore higher doses may be necessary for adequate CSF penetration [91,92].
Bedaquiline	Bedaquiline penetrated freely into the CSF of adults under treatment with pulmonary tuberculosis [94].
Clofazamine	Poor CSF penetration, which may be improved by chemical modification [95].
Cycloserine	Good CSF penetration of inflamed meninges [92,96].
Ethionamide	Good CSF penetration [90].
Delamanid	Very limited clinical data available, low total CSF levels [97].
Beta-lactamase inhibitors	
Avibactam	No data available.
Clavulanic acid	Very limited data suggest that amoxicillin-clavulanate may be effective for the treatment of bacterial meningitis [98,99].
Sulbactam	Very high CSF:plasma concentrations in combination with ampicillin [91]. However, clinical experience with this agent for meningitis is limited.
Tazobactam	No clinical data available.
Vaborbactam	No clinical data available.
Carbapenems	
Doripenem	No clinical data available.
Ertapenem	No clinical data available.
Imipenem	Measurable CSF penetrations, but high proconvulsive activity may restrict its use [100].
Meropenem	CSF concentrations adequate for treating meningitis [91].
Cephalosporins	
Cephalexin	Usually ineffective due to lower CSF:serum concentrations [91].
Cefazolin	CSF concentrations of uninflamed meninges close to the MIC of moderately susceptible bacteria [90].
Cefadroxil	Usually ineffective due to lower CSF:serum concentrations [91].
Cefaclor	No clinical data available.
Cefotetan	No clinical data available.
Cefoxitin	No clinical data available.
Cefprozil	No clinical data available.
Cefuroxime	Reaches CSF concentrations in excess of MIC [91].
Cephamycin	No clinical data available.
Cefdinir	No clinical data available.
Cefepime	Adequate CSF penetration for treatment of meningitis [90].
Cefixime	Cefixime crosses the blood brain barrier of inflamed meninges, but at limited concentrations and should therefore not be used to treat meningitis [91].
Cefotaxime	Adequate CSF penetration [90,91]
Ceftriaxone	Ceftriaxone has an adequate CSF penetration of inflamed meninges. CSF concentrations are lower compared with cefotaxime, most likely given the higher degree of protein binding of ceftriaxone. Nevertheless, ceftriaxone is an adequate agent for treatment of meningitis [90,91].
Ceftaroline	Different case studies reported that ceftaroline attained CSF concentration above MIC [101,102,103].
Ceftazidime	CSF attains therapeutic levels in CSF [90,91].
Ceftizoxime	Limited clinical data available suggest that ceftizoxime penetrates CSF [91].
Ceftobiprole	No clinical data available, clinical study ongoing (NCT04178629).
Cefiderocol	Very limited clinical data available in humans suggests that cefiderocol CSF concentrations in meningitis exceed MIC of gram negative organisms [104].
Fluoroquinolones	
Ciprofloxacin Delafloxacin Gatifloxacin Gemifloxacin Levofloxacin Moxifloxacin Norfloxacin Ofloxacin	As a group, fluoroquinolones demonstrate excellent CSF penetration. Clinical data are only available for ciprofloxacin, ofloxacin, levofloxacin and moxifloxacin [90].
Glycopeptides	
Teicoplanin	The high protein binding of teicoplanin restricts CSF penetration after IV administration [100].
Vancomycin	Vancomycin is highly hydrophilic and may reach sub therapeutic CSF concentration at conventional doses, but adequate concentrations at increased doses [91].
Dalbavancin	No clinical data available.
Telavancin	No clinical data available.
Glycylcycline	
Tigecycline	Limited clinical data available suggest that tigecycline reaches adequate concentrations of inflamed meninges [90].
Lincosamides	
Clindamycin Lincomycin	Lincomycin and its derivative Clindamycin is considered to have poor CSF penetration [91].
Monobactams	
Aztreonam	Scant clinical data available suggest that aztreonam reaches sufficient CSF concentrations after systemic administration in inflamed meninges [45].
Macrolides	
Azithromycin Clarithromycin Erythromycin Fidaxomicin	Macrolides have been unable to reach therapeutic CSF concentrations in adults [91].
Nitroimidazoles	
Metronidazole	Good CSF penetration in both inflamed and no inflamed meninges [90,91].
Tinidazole	No clinical data available.
Oxazolidinones	
Linezolid	CSF concentrations above the MIC of susceptible pathogens both with inflamed and uninflamed meninges [90].
Tedizolid	No clinical data available.
Penicillins	
Penicillin G Penicillin V	Good CSF concentrations after intravenous administration [91].
Temocillin	Very limited clinical data available suggest that temocillin may reach therapeutic concentrations in the CSF of patients with gram negative meningitis, but more data are necessary to assess this [105].
Amoxicillin Ampicillin	Good CSF penetrations after IV administration [91].
Cloxacillin	Penetrates in CSF of inflamed meninges to a limited extent, therefore higher doses may be necessary to attain therapeutic targets [106]. Furthermore, therapy failure has been described in patients under treatment for Staphylococcus meningitis [107].
Flucloxacillin	Penetrates in CSF of inflamed meninges to a limited extent, therefore higher doses may be necessary to attain therapeutic targets [106].
Nafcillin	Insufficient CSF penetration for treatment of meningitis [91].
Oxacillin	Limited CSF diffusion at conventional doses [108].
Piperacillin	Crosses the inflamed and non-inflamed blood-brain barrier but in unpredictable amounts [109].
Ticarcillins	Very limited data available, rather low and variable CSF concentrations after administration of ticarcillin-clavulanate [110].
Polymyxins	
Polymyxin B Polymyxin E (Colistin)	Limited clinical data available suggest very low CSF penetration after systemic administration [111].
Sulfonamides	
Sulfamethoxazole	High doses achieve good CSF concentrations both with inflamed and uninflamed meninges [90].
Tetracyclines	
Doxycycline	Limited clinical data, same CSF penetration in both inflamed and uninflamed meninges [90].
Minocycline	No clinical data available.
Miscellaneous	
Chloramphenicol	Chloramphenicol penetrates well into CSF, but significant toxicities prohibit the clinical use [90,91].
Daptomycin	Limited PK data available on CSF penetration. Some case reports described the successful use of daptomycin in meningitis.
Fosfomycin	Enters the CSF in the presence and absence of meningeal inflammation [90].

### 4.3. Metabolism

Metabolism is the process of chemical modification of a drug molecule (substrate) into a hydrophilic metabolite [112]. This is typically needed to eliminate lipophilic drugs, as these do not dissolve in water and therefore preclude renal excretion [113]. Some metabolites are toxic, other substrates only become active after metabolism; these substrates are therefore called pro-drugs.

Drug metabolism is divided in two main phase reactions [114]. Phase I reactions introduce a functional group to the substrate through oxidation, dealkylation, reduction or hydrolysis; therefore, phase 1 reactions are referred to as functionalization. Importantly, substrates can remain pharmacologically active after phase 1 reactions. Phase II reactions inactivate a substrate through adding a polar conjugate, most commonly glucuronide, which facilitates subsequent elimination through urine or bile. The extent of metabolism of a molecule is determined by the molecular structure, as a substrate may undergo either or both phase reactions [115].

Although most metabolism happens in the liver, other sites of metabolism include the kidneys, lung, intestines, brain and muscle [116]. Cytochrome P450 (CYP) is the major enzyme family responsible for phase 1 reactions [117]. Cytochromes are a superfamily of proteins containing heme as a cofactor, with the role of enzymatic metabolism of both endogenous substrates such as steroids or lipids, and of exogenous substrates such as nutrients and drugs [118].

CYP isoenzymes activity changes during lifetime, increasing significantly during the early years of life, when activities ultimately become similar to adult levels [119]. Therefore, lower doses of drugs that require hepatic metabolism may be needed during the early years of life, while a pro-drug can have lower efficacy early in life. To make drug metabolism even more complex, some isoenzymes like CYP3A7 may be more expressed during the early weeks of life or even in the fetus [120]. Furthermore, phase II enzymes may have varying expression during different periods of life. For example, uridine 5′-diphosphoglucuronic acid glucuronosyltransferases (UGT), which is responsible for about 15% of drug metabolism, is less expressed during early weeks of life. Exemplarily, the grey baby syndrome was observed in infants who were treated with the antimicrobial chloramphenicol, as their low UGT activity restricted metabolism, resulting in mitochondrial chloramphenicol toxicity [121].

Clinically relevant drug interactions may occur in the metabolism processes. Drug interactions are best understood by examining the three main actors: (i) the substrate drug, which is usually metabolized by CYP enzymes, (ii) the inducer drug that can increase the synthesis of CYP enzymes and potentially increase the metabolism of a substrate (thus decreasing serum concentrations of the substrate); (iii) the inhibitor drug that can inhibit CYP and potentially decrease the metabolism of a substrate (increasing serum concentrations of the substrate) [122]. For example, rifampicin is a potent CYP inducer that leads to reduced concentrations of substrate, causing subtherapeutic concentrations [123]. On the contrary, CYP inhibitors such as macrolides and isoniazid increase concentrations of the substrate, therefore increasing the chance of toxicity [124]. An important concept is that inhibition processes require hours and therefore has a relatively quick effect on the substrate’s drug concentration, while induction events require nuclear transcriptional effects that take days to weeks [122].

### 4.4. Elimination

Elimination or excretion (also called clearance, CL) represents the process of removal of drugs or by-products from the body, sometimes following metabolism [115]. The renal and hepatic routes are the most common ways of elimination, other routes include the lungs, intestine and secretory glands such as sweat, saliva and tears. Half-life (t_1/2_) is a pharmacokinetic parameter of elimination, which is defined as the time to reduce C_max_ by 50%. t_1/2_ is dependent on both V_d_ and clearance, as is mathematically expressed by the formula t_1/2_ = 0.693 × V_d_/CL [125].

The renal route is the commonest form of eliminating antimicrobials [126] with glomerular filtration rate (GFR) as its major determinant, which is dependent on renal blood flow. On top of GFR, the tubular processes secretion and reabsorption also can be of relevance for elimination [127]. Both maturational and non-maturational factors affect the performance of renal clearance throughout childhood. The maturational changes associated with renal clearance are related to increases in glomerular filtration rate and tubular secretion with age. Glomerular filtration starting from 2 mL/min in a newborn markedly increases in the first year of life, reaching adult rates by 1–2 years of age. Age-related tubular secretion changes occur in the form of an increase in number and isoforms of transporters mediated by an increase in serum glucocorticoids and thyroid hormone, occurring in synchrony with weaning, and show notable increments after five years of age [128].

Multiple non-maturational factors affect renal excretion in children. However, factitious changes should be discerned. For example, trimethoprim competitively interferes with the tubular secretion of creatinine. Therefore, prolonged administration of trimethoprim results in an increase in serum creatinine, without a decreased GFR, due to impaired tubular secretion [129]. Hence, while trimethoprim has a potential to cause nephrotoxicity, isolated increases in serum creatinine should not be taken as an indicator of renal injury. Disease states are also relevant non-maturational factors affecting antimicrobial clearance in children in both directions, as both an increase (hyperfiltration) or decrease (renal impairment) can occur. Augmented renal clearance triggers low plasma concentrations of administered antimicrobials [129]. High cardiac output and the subsequent raised glomerular filtration (>10% increase from normal clearance rates) are associated with the development of augmented renal clearance [130]. Infants and children, especially those in the post-traumatic or post-operative period, having sepsis, burns or hematologic malignancies are at a higher risk of ARC [131,132]. In order to maintain adequate drug exposure in such children, either prolonged infusions, frequent dosing and increased dosing or change to an alternative antimicrobial drug are required [133,134,135].

## 5. Therapeutic Drug Monitoring

Therapeutic drug monitoring, which refers to individual patient dose adjustments based on measured drug concentrations, is an important tool to optimize therapy. It may be indicated in patients where high peak concentrations are desired in infections caused by organisms with high MICs, or in patients who are receiving antimicrobials that exert dose-dependent toxicities such as nephrotoxicity. In the case of aminoglycosides, which are nephrotoxic drugs, monitoring through concentrations could limit toxicity. Additionally in cases of treating organisms with higher MICs there may be a need to achieve higher peak concentrations since C_max_/MIC is the PK/PD parameter of efficacy associated with aminoglycosides. Furthermore, pharmacokinetic variability exists with many drugs, this compounded with physiological and anatomical changes seen in pediatric patients further supports the need for therapeutic drug monitoring. Many studies have recommended therapeutic drug monitoring to address the pharmacokinetic variability seen with antituberculosis agents [136,137,138]. Newer drug monitoring approaches are aimed at achieving targeted AUC as opposed to single trough concentrations. Such practices are apt for drugs that utilize AUC/MIC as a PK/PD parameter, such as vancomycin. Newer vancomycin guidelines have proposed targeted therapy by achieving appropriate AUC [139]. Model-based approaches have allowed for easier computing of the AUC for individualization of therapy but requires a drug concentration for dose prediction [139]. Notably, Ewoldt et al. in a recent study reported no beneficial effect of model informed precision dosing of beta-lactam antibiotics and ciprofloxacin on ICU length of stay in critically ill patients however, the study was conducted in adult patients and did not include the pediatric population [140]. In the case of aminoglycosides, which are nephrotoxic drugs, monitoring trough concentrations during the treatment course could limit toxicity. Additionally in cases of treating organisms with higher MICs there may be a need to achieve higher peak concentrations since C_max_/MIC is the PK/PD parameter of efficacy associated with aminoglycosides.

## 6. Challenges in Attaining Effective Drug Concentrations in Children Living in LMIC

Just under half (45%) of all deaths in children under 5 years of age globally are attributed to undernutrition. Infections are responsible for a large proportion of mortality in malnourished children, due to reduced serum immunoglobulin levels, atrophied immune tissues like the thymus, and impaired epithelial mucosal barriers, such as the skin, respiratory and intestinal tract [141,142]. Therefore, the World Health Organization recommends that all children admitted with severe acute malnutrition receive broad-spectrum antimicrobials. However, different comorbidities and restrictions in resources make it challenging to attain PK-PD targets in LMIC settings. There are relatively few studies available that investigated the effects of protein-energy malnutrition on the pharmacokinetics of medicines in children [143]. While a complete overview of the effects of malnutrition on antimicrobial pharmacology is beyond the scope of this review, it is important for clinicians to recognize that the severity of malnutrition and characteristics of the drug will impact pharmacokinetics. In general, the volume of distribution and drug disposition will be increased for hydrophilic drugs and decreased for lipophilic drugs in severe malnutrition [144]. In addition, alterations in total body water, muscle mass and serum protein concentrations in malnourished children will further impact drug delivery [145]. Dose modifications of enteral medications may be necessary to account for impaired absorption and the reduced total drug clearance associated with malnourishment [146]. Furthermore, hypoalbuminemia associated with severe acute malnutrition can result in hydrophilic drugs having larger volumes of distribution. Therefore, higher doses may be necessary to attain adequate serum concentrations to treat blood stream infections [147,148].

### 6.1. Co-Morbidities

Apart from the effects of malnutrition, many diseases in LMIC may alter the pharmacokinetics of antimicrobials. The faster intestinal transport of oral antimicrobials during prolonged diarrhea leads to reduced absorption [149]. Furthermore, HIV and tuberculosis are common diseases in LMIC that both require drugs affecting P-gp. Rifampicin is an inducer, while protease inhibitors are either inhibitors or substrates of P-gp, which potentially results in drug-drug interactions [150]. Augmented renal clearance and chronic kidney disease are commonly observed in children with sickle cell disease that may warrant dose modifications to attain therapeutic targets and avoid toxicity [151,152]. Causes of childhood kidney disease in developing countries are diverse and mainly relate to antecedent post streptococcal glomerulonephritis, dehydration, malaria, use of herbal medicines and lead to various syndromes like hemolytic uremic syndrome, acute tubular necrosis and glomerulonephritis. A full understanding of the spectrum of etiologies is hampered by limited diagnostics [153,154]. A reduced creatinine clearance was also observed in children living with HIV in a population pharmacokinetic study of levofloxacin among South African children receiving treatment for multi-drug resistant tuberculosis [155]. Renal dysfunction associated with HIV infection is driven by direct renal parenchymal infection and immune-complex deposition [156].

### 6.2. Altered Polymorphisms

Besides disease related alterations, polymorphisms in drug metabolizing enzymes may further affect the pharmacokinetics of antimicrobials in affected patients. As an example, isoniazid is included in both prophylactic and therapeutic regimens against tuberculosis while it is widely appreciated that this has potential severe side effects. Its metabolism is highly dependent on the individual acetylation profile of the *N-acetyltransferase* (*NAT2*) gene. There is robust evidence that NAT-related polymorphisms already impact isoniazid clearance from neonatal life onwards, as the metabolic activity increases steadily from 4 months until 17 years of age (r = 0.53, age range 4 months to 17 years, 25/88 cases were 4–23 months) [157]. Slow genotypes (no alleles) had a much lower metabolic ratio compared to rapid (two alleles) genotypes (2-fold difference). Schaaf et al. estimated the first order elimination rate constant in 64 children [158], which related both to age and NAT-2 allele frequency (SS = 0.254; FS = 0.51; FF = 0653 h^−1^). Finally, Zhu et al. quantified the pharmacogenetic specific NAT2 enzyme maturation in perinatal HIV exposed infants receiving isoniazid [159]. Consecutive plasma concentration-time measurements of isoniazid from 151 infants (starting at 3–4 months of age) receiving isoniazid 10 to 20 mg/kg/day orally during the 24-month study were incorporated in a population analysis along with NAT2 genotype, body weight, age, and sex. For fast (FF) and intermediate (SF) acetylators, clearance increased from 14.25 L/h. 70 kg and 10.88 L/h. 70 kg at 3 months to 22.84 L/h. 70 kg and 15.58 L/h. 70 kg at 24 months, while slow (SS) acetylators displayed no changes over age (7.35 L/h). Comparing slow to fast acetylators, there is a 2-fold difference at 3 months, to further increase to a 3-fold difference at 24 months. How to implement such information to attain a ‘precision approach’ within a LMIC remains an issue, but perhaps awareness and considering concentration guided dosing shortly after initiation could be a way forward.

### 6.3. Challenges with Healthcare Administration

Administrative challenges in hospitals in LMIC include work over-load on behalf of pediatric nurses, who often work in crowded wards, which potentially leads to errors in dosing. Furthermore, skipped doses are often-unnoticed [160]. Under-dosing or inappropriate frequency of antimicrobial dosing will inevitably promote AMR [161]. Limitations in therapeutic drug monitoring in many low-income nations need to be overcome to combat in particular resistance to glycopeptide and aminoglycoside antibiotics [162,163]. LMIC face many challenges due to limited resources and the cost associated with effectively running such facilities. Due to the rural settings of most clinics and hospitals, samples need to be transported over vast distances to reach therapeutic drug monitoring laboratories if one is available, adequate storage of samples are impeded by lack of equipment and conveying of results may be delayed due to the lack of network signal. Addressing these issues would require larger stakeholder engagement with governmental involvement to set up infrastructure, adequately train staff in conducting and interpretation of therapeutic drug monitoring, and devise ways to minimize high-cost burden such as monitoring in selected patients with limited sampling and the use of model-based precision dosing. In the long term, adequate monitoring practices could off-set the cost associated with lack of drug efficacy and/or drug toxicity in the absence of adequate monitoring.

Overall, the practice of infectious diseases and clinical microbiology is hampered by limited clinical bacteriology laboratories, shortage of pharmacokinetic and pharmacodynamics studies as well as lack of access to therapeutic drug monitoring [162,164].

## 7. Conclusions

A thorough understanding of developmental pharmacokinetics is pivotal for adequate dosing in children. In this review, we outlined the effects of growth and maturation on pharmacokinetics in children. Subsequently, we elaborated how common comorbidities, such as malnourishment, co-morbidities such as tuberculosis, augmented renal clearance, HIV and tuberculosis, in LMIC may further affect the pharmacokinetics of antimicrobials in children. Limited resources for therapeutic drug monitoring restrict the abilities to individualize doses based on measured concentrations. Further pharmacokinetic studies of antimicrobials in children in LMIC are urgently needed to optimize dosing, and hence to attain PK/PD targets and combat antimicrobial resistance.

## Figures and Tables

**Figure 1 antibiotics-12-00017-f001:**
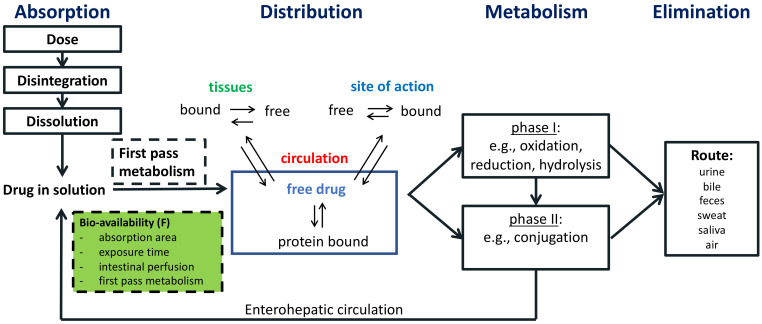
Overview of the pharmacokinetic processes absorption, distribution, metabolism and elimination.

**Figure 2 antibiotics-12-00017-f002:**
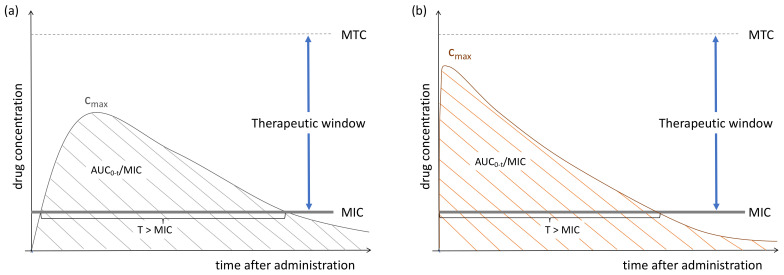
Concentration-time curve following oral administration or prolonged infusion (**a**) and after intravenous administration (**b**). Abbreviations: AUC_0-t4_: area under the curve between dosing interval. C_max_: maximum concentration. T: time. MIC: minimum inhibitory concentration. MTC: minimum toxic concentration.

**Table 1 antibiotics-12-00017-t001:** (A) Physicochemical & pharmacokinetic characteristics of antimicrobials mainly eliminated by urine. (B) Physicochemical and pharmacokinetic characteristics of antimicrobials mainly eliminated by feces. (C) Physicochemical and pharmacokinetic characteristics of antimicrobials mainly eliminated by bile.

(A)								
**Class**	Agent	PK/PD Index	Molecular Weight (g/mol) [20]	pKa [21]	LogP [20]	Fraction Protein Binding (%) [20]	Metabolism [21]	Alternative Route of Elimination [21]
Aminoglycosides	Amikacin	C_max_/MIC [22]	585.6	8.1–12.1	−8.8–−7.4	<10%	Aminoglycosides are not significantly metabolized.	
	Gentamicin		477.6	10.1–12.6	−4.1–−1.9	0–30%		
	Kanamycin		484.5	9.5–12.1	−6.9–−6.3	N/A		
	Neomycin		614.6	12.9 [23]	−9–−3.7	N/A		
	Streptomycin		581.6	11.1–11.6	−8–−2.5	N/A		
	Spectinomycin		332.4	7.0–9.2	−3.1–−2.3	Not significant		
	Tobramycin		467.5	9.7–12.5	−6.2–−5.8	Not significant		
First line anti- mycobacterials	Isoniazid	AUC/MIC [24]	137.1	1.8–13.6	−0.8–−0.7	0–10%	Hepatic	
	Pyrazinamide	AUC/MIC [24]	123.1	−0.5–13	−1–−0.6	~10%	Mainly hepatic	
	Rifabutin	AUC/MIC, C_max_/MIC [25]	847.0	6.9–9.0	4.1–4.7	85%	Hepatic	Feces
Second line antimycobacterials	Cycloserine	T > MIC [26]	102.1	4.2–8.4	−1.5–−0.9	N/A	Hepatic [27]	
	Ethionamide	AUC/MIC [28]	166.3	5–11.9	0.4–1.1	~30%	Extensive hepatic metabolism	
Beta-lactamase inhibitors	Clavulanic acid	T > MIC [29]	199.2	−2.6–3.2	−2.3–−1.2	~25% for amoxicillin-clavulanic acid	Hepatic	Feces, exhaled air
	Sulbactam		233.2	−3.8–3.1	−1	~38%	<25% is metabolized by the liver [30]	
	Tazobactam		300.3	0.8–2.9	−2	~30%	Hepatic	
	Avibactam		265.3	−3.9–−2	−1.8	5.7–8.2%	Not significant	
	Vaborbactam		297.1	−2.6–3.8	1.0–1.9 [31]	~33%	Not significant	
	Relebactam		348.4	−2–10	−3.6	~22%	Not significant	
Carbapenems	Doripenem	T > MIC [22]	420.5	3.3–9.5	−5.6, −1.3 [32]	8.1%	Limited hepatic metabolism	
	Ertapenem		475.5	3.2–9.0	0.3–1.5	85–95%	Limited hepatic metabolism	
	Imipenem		299.4	3.2–10.9	−0.7	20%	Renal metabolism	
	Meropenem		383.5	3.3–9.4	−2.4–−0.6	~2%	<30% of a dose undergoes hepatic metabolism	
First generation cephalosporins	Cephalexin	T > MIC [33,34]	347.4	3.3–7.2	0.6–0.7	10–15%	Not significant	
	Cefazolin		454.5	0.3–2.8	−0.6	74–86%	Not significant	
	Cefadroxil		363.4	3.3–7.2	−2.1–−0.4	28.1%	Not significant	
Second generation cephalosporins	Cefaclor		367.8	2.8–7.2	−2.3–0.9	23.5%	Not significant	
	Cefuroxime		424.4	−1.1–3.0	−0.8–−0.2	50%	Not significant	
	Cefuroxime axetil		510.5	−1.2–10.9	0.9	28–38% [35]	Axetil is metabolized by the liver	
	Cefotetan		575.6	−1.5–3.0	0.1	88%	Not significant	
	Cefoxitin		427.5	−3.8–3.4	0	31–54% [36]	Minimal hepatic metabolism	
	Cefprozil		389.4	3.3–7.2	−1.4–0.6	36%	Not significant	
	Cefmetazole		471.5	−1.7–3.2	−2.2–−0.6	85% [37]	Not significant	
Third generation cephalosporins	Cefdinir		395.4	2.7–9.7	−3.5–0	60–70%	Not significant	
	Cefditoren		506.6	2.3–3.7	0.7	88%	Not significant	
	Cefixime		453.5	2.5–4.0	−0.7–−0.4	65%	Hepatic	
	Cefpodoxime		427.5	2.8–3.6	−1.4	21–33%	Minimal hepatic metabolism	
	Ceftazidime		546.6	2.4–4.0	−1.6–0.4	5–23%	Not significant	
	Ceftizoxime		383.4	2.7–3.6	0	30%	Not significant	
	Ceftibuten		410.4	2.9–4.7	−0.3	65%	~10% is metabolized by the liver	
	Ceftriaxone		554.6	2.7–3.4	−1.7–−1.3	95%	Negligible	Bile
	Cefotaxime		455.5	2.7–3.6	−1.4–−0.5	8–41% [36]	Partially (15–20%) by the liver [38]	
	Ceftolozane		666.7	2.5–9.1	−6.2–−3.2	16–21%	Not significant	
Fourth generation cephalosporins	Cefepime		480.6	2.8–3.6	−0.1	20%	<1% is metabolized by the liver	
Fifth generation cephalosporins	Ceftobiprole		534.6	2.9–10.4	−2.4	<16% [39]	Minimal hepatic metabolism [39]	
	Ceftaroline		684.7	0.4–1.8	2.3	~20%	Minimal hepatic metabolism	Feces
Siderophore cephalosporins	Cefiderocol		752.2	2.6–4.0	−2.3–1	40–60%	Minimal hepatic metabolism	
Fluoroquinolones	Ciprofloxacin	AUC/MIC [34]	331.3	5.6–8.8	−1.1–2.3	20–40%	Up to 15% hepatic metabolism	Feces
	Delafloxacin		440.8	−1.3–5.6	2.7	84%	Hepatic	Feces
	Gatifloxacin		375.4	5.5–8.8	−0.7–2.6	20%	Limited hepatic metabolism	
	Levofloxacin		361.4	5.4–6.7	−0.4–2.1	24–38%	Very limited metabolism	Feces
	Norfloxacin		319.3	5.6–8.8	−1.0–2.1	10–15%	Hepatic and renal	Feces
	Ofloxacin		361.4	5.4–6.7	−0.4–2.1	32%	Hepatic	Feces
	Prulifloxacin		461.5	5.2–6.0	1.0	41–59% [40]	Hepatic	Feces
Glycopeptides	Teicoplanin	AUC/MIC [41]	1879.7	3.0–7.1	0.5	90–95% [42]	Minimal hepatic metabolism	
	Vancomycin		1449.3	3.0–9.9	−3.1–−2.6	~50%	Not significant	
Lipoglyco- peptides	Dalbavancin		1816.7	1.7–9.9 [43]	3.8	93%	Unlikely to have significant metabolism	Feces
	Telavancin		1755.6	1.6–10.0	−2.1	>90%	Unknown	
	Oritavancin		1793.1	2.2–10.0	1.5–4.1	85%	Not significant	Feces
Lincosamides	Clindamycin	AUC/MIC [22,34]	425.0	7.6–12.4	2.2	60–94% [44]	Hepatic	Feces
	Lincomycin		406.5	8.0–12.4	0.2–0.6	28–86%	Hepatic	Bile
Monobactams	Aztreonam	T > MIC [45]	435.4	−1.5–3.9	0.3	43–56%	6–16% is metabolized by the liver	
Nitroimidazoles	Metronidazole	AUC/MIC, C_max_/MIC [22,46]	171.2	2.6–15.4	−0.1–0	<20%	Hepatic	Feces
	Secnidazole	Undefined	185.2	3.1–15.2	0.2	<5–15%	N/A	
	Tinidazole	Undefined	247.2	3.3	−0.4–0.7	12%	Hepatic	Feces
Oxazolidones	Linezolid	AUC/MIC [22]	337.4	−1.2–14.9	0.7–1.3	~31%	Hepatic	
Natural penicillins	Penicillin G	T > MIC [22]	334.4	−2.8–3.5	1.5–1.8	45–68%	Hepatic	Bile
Aminopenicillins	Amoxicillin		365.4	3.2–7.2	−2–0.9	17%	Hepatic	
	Ampicillin		349.4	3.2–7.2	−1.1–1.4	8–25% [47]	Hepatic	
Semi-synthetic penicillins	Cloxacillin		435.9	−0.4–3.8	2.4–3	~94%	Intestinal	Bile
	Dicloxacillin		470.3	−0.7–3.8	2.9–3.7	96–97% [48]	Hepatic	
	Flucloxacillin		453.9	−0.9–3.8	2.6–3.2	95–96% [48]	Hepatic	
	Oxacillin		401.4	−0.1–3.8	2.4	92–96%	45–50% hepatic [49]	
	Temocillin [50]		414.5	−4.3–3.1	1.1	~80% [48]	N/A	
Ureidopenicillins	Piperacillin		517.6	−4.3–3.5	0.3–0.5	39.4–71.3% [51]	Not significant	Bile
Carboxy- penicillins	Ticarcillin		384.4	−6.3–3.1	0.8	45%	N/A	
Polymixins *	Polymyxin B	AUC/MIC [33]	1203.5	8.9–11.6	−2.5	79–92%	N/A	
Sulfonamides	Sulfadiazine	C_max_/MIC, AUC/MIC [22]	250.3	2.0–6.4	−0.2–−0.1	20–25% [52]	Hepatic	
	Sulfadoxine		310.3	3.4–6.1	0.7	~94% [53]	Hepatic	
	Sulfamethoxazole		253.3	2.0–6.2	0.7–0.9	~70%	Hepatic	
Tetracyclines	Doxycycline	AUC/MIC [54]	444.4	3.1–8.3	−0.7–0.6	>90%	Hepatic	Feces
	Tetracycline		444.4	3.3–9.3	−2–−1.3	20–67%	Not significant	Feces
Miscellaneous	Chloramphenicol	C_max_/MIC, AUC/MIC [22]	323.1	−2.8–8.7	0.7–1.1	50–60% in adults, 32% in premature neonates	Extensive hepatic metabolism	
	Daptomycin	AUC/MIC [22]	1619.7	3.0–9.6	−5.1	90–94% [55]	Minimum extent, metabolism site unknown [56]	Feces
	Fosfomycin	AUC/MIC [34]	138.1	−4.3–1.3	−1.6–−1.4	No plasma binding	Not significant	
	Trimethoprim	C_max_/MIC, AUC/MIC [22]	290.3	7.1–17.3	0.6–0.9	44%	Hepatic	
	Nitrofurantoin	Undefined	238.2	−2.2–8.3	−0.5	<90%	Hepatic	
(B)								
Class	Agent	PK/PD Index	Molecular Weight (g/mol) [20]	pKa [21]	LogP [20]	Fraction Protein Binding (%) [20]	Metabolism [21]	Alternative Route of Elimination [21]
First line anti- mycobacterials	Ethambutol	C_max_/MIC, AUC/MIC [24]	204.3	9.7–14.8	−0.4–0.4	20–30%	Hepatic	Urine
	Rifabutin	AUC/MIC, C_max_/MIC [25]	847.0	6.9–9.0	4.1–4.7	85%	Hepatic	Urine
Third line antimycobacterials	Bedaquiline	AUC/MIC, C_max_/MIC [57]	555.5	8.9–13.6	7.7	>99.9%	Hepatic	
	Clofazimine	Not identified [26]	473.4	6.6–16.2	7–7.7	N/A	N/A	
	Delamanid	Not Identified [26]	534.5	5.5	5.6	>99.5%	Hepatic	
Beta-lactamase inhibitors	Clavulanic acid	T > MIC [29]	199.2	−2.6–3.2	−2.3–−1.2	~25% for amoxicillin-clavulanic acid	Significant hepatic metabolism	Urine, exhaled air
Fluoroquinolones	Gemifloxacin	AUC/MIC [34]	389.4	5.4–9.4	−0.7–2.3	60–70%	Limited hepatic metabolism	Urine
	Moxifloxacin		401.4	5.5–9.5	0.6–2.9	50%	<50% hepatic metabolism	Urine
Macrolides	Clarithromycin	AUC/MIC, T > MIC [34]	748.0	9–12.5	1.7–3.2	~70%	Hepatic	Urine
	Fidaxomicin		1058	−1.4–5.9	6.4	31% [48]	Intestinal	
Oxazolidones	Tedizolid	AUC/MIC [22]	370.3	−1.7–14.6	1.4	70–90%	Hepatic	Urine
Tetracyclines	Eravacycline	AUC/MIC [54]	558.6	3.0–9.0	1	79–90%	Hepatic	Urine
	Omadacycline		556.6	2.9–10.5	3	~20%	Not significant	Urine
	Tigecycline		585.7	3.2–9.0	−0.2–1.1	71–89%	Hepatic [58]	Urine
	Daptomycin	AUC/MIC [22]	1619.7	3.0–9.6	−5.1	90–94% [55]	Metabolism site unknown [56]	Urine
(C)								
Class	Agent	PK/PD Index	Molecular Weight (g/mol) [20]	pKa [21]	LogP [20]	Fraction Protein Binding (%) [20]	Metabolism [21]	Alternative Route of Elimination [21]
First line Antimyco- bacterials	Rifampicin	AUC/MIC, C_max_/MIC [25]	822.9	1.7–7.4	2.7–4.9	90%	Hepatic	Urine
Third generation cephalosporins	Cefoperazone	T > MIC [33,34]	645.7	−1.7–3.2	−0.7	82–93%	Not significant	
Lincosamides	Lincomycin	AUC/MIC [22,34]	406.5	8.0–12.4	0.2–0.6	28–86%	Hepatic	Urine
Macrolides	Azithromycin	AUC/MIC, T > MIC [34]	749.0	8.5–12.4	3.0–4.0	7–51%	Hepatic	Urine
	Erythromycin		733.9	9–12.5	2.6–3.1	70–93%	Hepatic	Urine
Natural penicillins	Penicillin V	T > MIC [22]	350.4	−4.9–3.4	1.4–2.1	50–80%	Hepatic	Urine
	Penicillin G		334.4	−2.8–3.5	1.5–1.8	45–68%	Hepatic	Urine
Semi-synthetic penicillins	Cloxacillin		435.9	−0.4–3.8	2.4–3	~94%	Intestinal	Urine
	Nafcillin		414.5	−1.9–3.3	2.9–3.3	88.4–91.4%	Hepatic	
Ureidopenicillins	Piperacillin		517.6	−4.3–3.5	0.3–0.5	39.4–71.3% [51]	Not significant	Urine
Tetracyclines	Minocycline	AUC/MIC [54]	457.5	3.2–8.8	−0.6–0.1	76%	Hepatic	Urine

* The route of elimination for Polymyxin is unknown. For molecular weight, LogP and fraction protein binding refer to reference [20] unless cited otherwise. For pKa, metabolism and main route of elimination refer to reference [21] unless cited otherwise.

**Table 2 antibiotics-12-00017-t002:** Summary of antimicrobial-food interactions.

Agent	Food Effect on Absorption
Amoxicillin	No effect of fasting status for infants, children and adults [69,70].
Amoxicillin/clavulanate	Concomitant food ingestion may enhance absorption and reduce gastric upset [71].
Ampicillin	Impaired when taken with food. Therefore, if administered PO, ampicillin should be administered 1 h before or 2 h after meals [70].
Azithromycin	Tablets and suspension present no food effect [69].
Cephalexin, cefadroxil, cefaclor, cefprozil, cefixime	Not affected by food intake [72].
Cefuroxime axetil	Absorption and dissolution into active form are improved when taken with food [73,74].
Ciprofloxacin	Impaired by dairy products, Ca^2+^ and Mg^2+^ supplements [75].
Metronidazole	Food may decrease the rate but not the extent of absorption. However, food may reduce gastric upset [76].
Rifampicin	Impaired when taken with food, therefore should be taken on an empty stomach [77].
Tetracycline	Impaired when taken with food or with divalent metal cations, such as Fe^+2^ and Ca^+2^ [78].

## Data Availability

Not applicable.

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
