# Peer review of "Pharmacokinetics of Antimicrobials in Children with Emphasis on Challenges Faced by Low and Middle Income Countries, a Clinical Review"

_antibiotics, 2022, doi:10.3390/antibiotics12010017_

Round 1

Reviewer 1 Report

An interesting narrative overview of PK of antimicrobials in children in low and middle income countries. The main part of the paper is a well written standard description of factors affecting PK. Table 1 is too long - I suggest dividing by route of elimination. I note that antimalarials are not included. Table 3 would benefit from exclusion of the drugs for which there are no data.

Section 5 deals with LMIC and would benefit from subheadings (malnutrition, coexisting infections, altered polymorphisms, and administration) and expansion. Access to medicines needs mentioning.

The section on malnutrition needs expanding. An old SR suggested limited data (see  Oshikoya KA, Sammons HM, Choonara I.  A systematic review of pharmacokinetics studies in children with protein-energy malnutrition.  Eur J Clin Pharmacol 2010; 66: 1025-1035.)

The effect of coinfections on PK needs mentioning.

It would be helpful to include antimalarials

Author Response

Thank you so much for reviewing our manuscript and for all your valuable comments.

An interesting narrative overview of PK of antimicrobials in children in low- and middle-income countries. The main part of the paper is a well written standard description of factors affecting PK.

Table 1 is too long - I suggest dividing by route of elimination. I note that antimalarials are not included.

This comment is well taken. As per your suggestion, we divided this per route of elimination. Antimalarials are indeed not included, since we aimed to provide a review of antimicrobials that are important in the treatment of bacterial infections, given the increasing antimicrobial resistance in LMIC. Certainly, malaria has remained a prevalent disease in LMIC, but we feel that including antimalarials within this manuscript would make the text too long. Therefore, we clarified the outline of our manuscript in lines 94-95.

Table 3 would benefit from exclusion of the drugs for which there are no data.

We agree that the table is long. Yet, meningitis is a common disease in LMIC, and choosing an antimicrobial with adequate CSF penetration is essential. Therefore, we elected to outline for which antimicrobials there are no data, rather than eliminating this.

Section 5 deals with LMIC and would benefit from subheadings (malnutrition, coexisting infections, altered polymorphisms, and administration) and expansion. Access to medicines needs mentioning.

Thanks for these suggestions. In the revised manuscript, we included subheadings and expanded the section as per your suggestion. Please note that this is now Section 6 in the revised manuscript.

The section on malnutrition needs expanding. An old SR suggested limited data (see  Oshikoya KA, Sammons HM, Choonara I.  A systematic review of pharmacokinetics studies in children with protein-energy malnutrition.  Eur J Clin Pharmacol 2010; 66: 1025-1035.)

We have added some additional details to the malnutrition section (i.e. introductory paragraph for Section 6) and have included this reference in the revision. Yet, given the complexities of how malnutrition may influence PK, we have not delved deeply into this topic. But, we provide additional references for the reader.

The effect of coinfections on PK needs mentioning.

Thank you for this suggestion. We believe that the addition of subheadings in Section 6 will emphasize the effect of co-infections, as well as other disease states, on PK.

Reviewer 2 Report

I appreciate the efforts and dedication about what authors have studied. The article is well written and easy to comprehend. Authors well reviewed an important issue - Pharmacokinetics of antimicrobials in children in low- and middle-income countries. I propose to accept the current version for publication.

Thank you.

Author Response

Thanks for reviewing our paper and thank you for this comment.

Reviewer 3 Report

Meesters et al. wrote a narrative review titled “Pharmacokinetics of antimicrobials in children in low- and middle-income countries, a clinical review”. The authors present an interesting manuscript about differences in pharmacokinetics between pediatric and adult patient populations, and address associated challenges faced in low- and middle-income countries. 

The authors provide an overview about the essentials of pharmacokinetics and include comprehensive tables depicting pharmacokinetic characteristics of various antimicrobials.

However, the article also has some shortcomings.

General

Although addressed in the title, the majority of the manuscript does not focus on pharmacokinetics of antimicrobials in children in LMIC. Rather, it mainly revolves around the general principles of PK/PD and PK in healthy pediatric populations. In addition, although well-structured and interesting, the section addressing CSF penetration, may be too elaborate and exceeds the primary aim of this review. Several reviews dedicated solely to these fields already exist. Therefore, I suggest shortening the background section, and expanding the final chapter.

E.g.: Line 311: What kinds of infections and comorbidities are responsible for a large proportion of mortality in malnourished children? How does malnutrition lead to infections? What are the typical illnesses associated with LMIC?

Line 314: What are the restrictions in resources and how do they make it challenging to attain PK-PD targets? 

Lines 366: The authors do not address the principles of TDM in their review. How is TDM being implemented in LMIC? What are the limitations in LMIC that need to be overcome? Would better TDM and target attainment lead to better outcomes? (In light of recent findings by Ewoldt et al. https://doi.org/10.1007/s00134-022-06921-9)  

The authors may also consider providing a table to summarize their findings

Consider revising the title. The PK of healthy children in LMICs is most likely similar to that in other regions. Rather, diseases and treatments associated with LMIC may influence their PK.

Please revise PK abbreviations throughout the manuscript. fT>MIC does not refer to the fraction of time, but rather the proportion (in %) of the time of a dosing interval in which the free concentration is above the MIC. As stated in the manuscript, only the free fraction of a drug may exert pharmacological effect and is largely dependent on a drug’s PPB. Therefore, please consistently address either all PK parameters using the free fraction abbreviation (fT>MIC), or without (T>MIC). Furthermore, The AUC interval of interest is not always 24h (AUC24), as it depends on the suggested dosing interval of each respective antibiotic. Instead, please use AUC0-t.

Figure 2: I suggest including the information/titles “oral administration or prolonged infusion” and “intravenous administration” in the figures themselves, rather than solely providing an explanation in the description. In addition, the authors may consider including the abbreviation “MTC”, as it is addressed in the body of the manuscript. The authors may consider visually linking the MIC and therapeutic window to their respective lines.

Table 1: The Column title “Class” is unsuitable, as the drug classes are described in a designated row. I suggest emphasizing drug classes (e.g. bold or larger text) for better legibility. Where no exact values are available, I suggest stating N/A, as opposed to wordings such as “Highly” or “Not completely understood”. In the section of the main route of elimination: If only <1% of Ethionamide is eliminated in urine, what is then the main route of elimination? What is meant with, “air” in the row of Clavulanic acid? Tetracycline has an inactive PMID. 

Minor Comments

Lines 147-149: This sentence feels out of place, I suggest starting with the physiological overview, followed by clinical examples. 

Line: 158: I suggest creating a new subchapter, when starting to address pediatric patients and their respective physiological differences. 

Line 175: I suggest creating a new subchapter to address the interaction between food and drug absorption. Consider moving this chapter to before “Withing 48 hours after birth…”

Line 215: As stated above, I believe the information on CSF penetration is too elaborate and exceeds the scope of the review. It is unclear if Table 3 addresses CSF penetration in children or adults. 

Line 293: This paragraph may be too elaborate, as it is not directly linked to children in LMIC.

Line 348: The results of Schaaf et al. are explained in great detail. I suggest shortening.

Conclusion: Limited access to TDM is not the greatest challenge in treating children in LMIC. In summary, what are the main challenges and how does this review contribute to overcoming these limitations?

Acknowledgments: It is unclear how these acknowledgments are associated with the present manuscript.

Author Response

Thanks so much for reviewing our manuscript and for your suggestions that will improve the quality of this review.

General

Although addressed in the title, the majority of the manuscript does not focus on pharmacokinetics of antimicrobials in children in LMIC. Rather, it mainly revolves around the general principles of PK/PD and PK in healthy pediatric populations.

This comment is well taken. We aimed to write an educational review that enhances the reader’s understanding of how comorbidities and environmental factors may alter PK/PD of antimicrobials. A good comprehension of general PK/PD principles in children is imminent for this, therefore we left this in the text.

In addition, although well-structured and interesting, the section addressing CSF penetration, may be too elaborate and exceeds the primary aim of this review. Several reviews dedicated solely to these fields already exist. Therefore, I suggest shortening the background section, and expanding the final chapter.

Indeed, there are different reviews on CSF penetration of antimicrobials, which we cite in our section. Yet, as CNS infections are common in LMIC and this often results in dilemmas around the optimal agent, we find it important to address this in our review.

E.g.: Line 311: What kinds of infections and comorbidities are responsible for a large proportion of mortality in malnourished children? How does malnutrition lead to infections? What are the typical illnesses associated with LMIC?

We elaborate on this in lines 404-408 of our revised manuscript.

Line 314: What are the restrictions in resources and how do they make it challenging to attain PK-PD targets?

We discuss this in lines 486-496 of the revision.

Lines 366: The authors do not address the principles of TDM in their review. How is TDM being implemented in LMIC? What are the limitations in LMIC that need to be overcome? Would better TDM and target attainment lead to better outcomes? (In light of recent findings by Ewoldt et al. https://doi.org/10.1007/s00134-022-06921-9)  

Thank you for this comment. Indeed, a general section on TDM was missing, we included this in the revision in section 5. Furthermore, we elaborate on the challenges on TDM in LMIC in section 6.

The authors may also consider providing a table to summarize their findings

Thanks for this suggestion. Yet, given the general concerns of length raised by other reviewers, we have opted not to add additional tables. We provided a summary in section 7.

Consider revising the title. The PK of healthy children in LMICs is most likely similar to that in other regions. Rather, diseases and treatments associated with LMIC may influence their PK.

We agree to this comment and have changed our title accordingly.

Please revise PK abbreviations throughout the manuscript. fT>MIC does not refer to the fraction of time, but rather the proportion (in %) of the time of a dosing interval in which the free concentration is above the MIC. As stated in the manuscript, only the free fraction of a drug may exert pharmacological effect and is largely dependent on a drug’s PPB. Therefore, please consistently address either all PK parameters using the free fraction abbreviation (fT>MIC), or without (T>MIC). Furthermore, The AUC interval of interest is not always 24h (AUC24), as it depends on the suggested dosing interval of each respective antibiotic. Instead, please use AUC0-t.

We revised all PK abbreviations in the manuscript.

Figure 2: I suggest including the information/titles “oral administration or prolonged infusion” and “intravenous administration” in the figures themselves, rather than solely providing an explanation in the description. In addition, the authors may consider including the abbreviation “MTC”, as it is addressed in the body of the manuscript. The authors may consider visually linking the MIC and therapeutic window to their respective lines.

We changed the lay-out of the figure and included MTC. Yet, as per MDPI’s instruction for authors, the title should be below the figure. We aimed to keep both figures separately for clarity reasons.

Table 1: The Column title “Class” is unsuitable, as the drug classes are described in a designated row. I suggest emphasizing drug classes (e.g. bold or larger text) for better legibility. Where no exact values are available, I suggest stating N/A, as opposed to wordings such as “Highly” or “Not completely understood”. In the section of the main route of elimination: If only <1% of Ethionamide is eliminated in urine, what is then the main route of elimination? What is meant with, “air” in the row of Clavulanic acid? Tetracycline has an inactive PMID. 

As per the feedback of reviewer I, we thoroughly revised table 1, omitted the column class and used bold text to emphasize on the classes. We chose to provide qualitative descriptions if exact values were unavailable, rather than stating that this information is unavailable. Antimicrobials that are (partly) eliminated via air refers to exhaled air, we clarified that in the table. Last, we omitted the inactive PMID for tetracycline.

Minor Comments

Lines 147-149: This sentence feels out of place, I suggest starting with the physiological overview, followed by clinical examples. 

We re-structured this section after your suggestions.

Line: 158: I suggest creating a new subchapter, when starting to address pediatric patients and their respective physiological differences. 

We re-structured this section after your suggestions.

Line 175: I suggest creating a new subchapter to address the interaction between food and drug absorption. Consider moving this chapter to before “Withing 48 hours after birth…”

We re-structured this section after your suggestions.

Line 215: As stated above, I believe the information on CSF penetration is too elaborate and exceeds the scope of the review. It is unclear if Table 3 addresses CSF penetration in children or adults. 

This comment is in line with the point that you raised earlier. We elected not to omit this section as CNS infections are common in LMIC.

Line 293: This paragraph may be too elaborate, as it is not directly linked to children in LMIC.

Renal physiology is a complex but imminent topic for a good understanding of renal elimination. Therefore, we have not shortened on this.

Line 348: The results of Schaaf et al. are explained in great detail. I suggest shortening.

Thanks for this suggestion, we shortened this sentence.

Conclusion: Limited access to TDM is not the greatest challenge in treating children in LMIC. In summary, what are the main challenges and how does this review contribute to overcoming these limitations?

We revised this section based on your comments.

Acknowledgments: It is unclear how these acknowledgments are associated with the present manuscript.

Thanks for this comment- we rephrased the acknowledgements in order to express the link with the current manuscript.

Round 2

Reviewer 3 Report

I have no further comments or suggestions.